# Ependyma in Neurodegenerative Diseases, Radiation-Induced Brain Injury and as a Therapeutic Target for Neurotrophic Factors

**DOI:** 10.3390/biom13050754

**Published:** 2023-04-27

**Authors:** Xin-Yu Ma, Ting-Ting Yang, Lian Liu, Xiao-Chun Peng, Feng Qian, Feng-Ru Tang

**Affiliations:** 1Department of Physiology, School of Basic Medicine, Health Science Center, Yangtze University, Jingzhou 434023, China; 2Department of Pharmacology, School of Basic Medicine, Health Science Center, Yangtze University, Jingzhou 434023, China; 3Department of Pathophysiology, School of Basic Medicine, Health Science Center, Yangtze University, Jingzhou 434023, China; 4Radiation Physiology Laboratory, Singapore Nuclear Research and Safety Initiative, National University of Singapore, Singapore 138602, Singapore

**Keywords:** neurodegenerative diseases, radiation-induced brain injury, ependyma, epidermal growth factor (EGF), brain-cerebrospinal fluid barrier, cerebrospinal fluid

## Abstract

The neuron loss caused by the progressive damage to the nervous system is proposed to be the main pathogenesis of neurodegenerative diseases. Ependyma is a layer of ciliated ependymal cells that participates in the formation of the brain-cerebrospinal fluid barrier (BCB). It functions to promotes the circulation of cerebrospinal fluid (CSF) and the material exchange between CSF and brain interstitial fluid. Radiation-induced brain injury (RIBI) shows obvious impairments of the blood–brain barrier (BBB). In the neuroinflammatory processes after acute brain injury, a large amount of complement proteins and infiltrated immune cells are circulated in the CSF to resist brain damage and promote substance exchange through the BCB. However, as the protective barrier lining the brain ventricles, the ependyma is extremely vulnerable to cytotoxic and cytolytic immune responses. When the ependyma is damaged, the integrity of BCB is destroyed, and the CSF flow and material exchange is affected, leading to brain microenvironment imbalance, which plays a vital role in the pathogenesis of neurodegenerative diseases. Epidermal growth factor (EGF) and other neurotrophic factors promote the differentiation and maturation of ependymal cells to maintain the integrity of the ependyma and the activity of ependymal cilia, and may have therapeutic potential in restoring the homeostasis of the brain microenvironment after RIBI or during the pathogenesis of neurodegenerative diseases.

## 1. Introduction

Neurodegenerative diseases are supposed to be caused by the abnormal structure and function of neurons [1]; they mainly include Alzheimer’s disease (AD) [2], Parkinson’s disease (PD) [3], amyotrophic lateral sclerosis (ALS) [4] and Huntington’s disease (HD) [5]. According to their clinical symptoms, neurodegenerative diseases can be classified into dementia and dyskinesia [6]. These neurodegenerative diseases not only seriously affect the quality of life of patients and their families but also bring great pressure to society. The total number of people with dementia has reached 55.2 million globally in 2019 and has been predicted to reach 78 million by 2030 and 139 million by 2050 [7]. Although efforts have been made to reveal the pathogenesis that leads to neurodegenerative diseases, the detailed mechanism underlying the structural and functional changes of neurons is still unclear [8]. As the most common neurodegenerative disease, AD is characterized by β-amyloid (Aβ) plaque and neurofibril tau entanglement [2]. Currently, updated data support that the pathogenesis of AD is initiated from dysregulated lipid metabolism, which contributes to the consequent myelination and blood–brain barrier (BBB) impairments [9,10]. Brain imaging studies in patients of AD and cerebral small vessel disease suggested that cerebrovascular dysfunction promoted neurodegenerative changes and the onset of cognitive decline [11,12]. In addition, some studies have shown that the pathogenesis of PD involves the abnormal accumulation of alpha-synuclein (α-syn), tryptophan metabolites and the consequent degeneration of the nigra-striatal pathway [13,14,15]. HD and ALS are genetic diseases associated with abnormal cytoplasmic protein aggregation and the subsequent neuroinflammation [4,5,16]. These previous studies suggest that the neurodegenerative diseases may be related to changes of the brain microenvironment.

Ionizing radiation has been applied to “open” the BBB for drug delivery [17] or anti-AD therapy [18]. This strategy has been updated by the magnetic-resonance-guided low-intensity focused ultrasound because of the reversibility of its BBB opening effect [19]. Recent research has emphasized the neuroinflammation and cognitive impairment after radiation-induced brain injury (RIBI) [20]. After the acute brain injury, the activated microglia and astroglia may recruit immune cell infiltration through the damaged BBB [21,22]. The inflammatory process in CSF can destroy the integrity of the choroid plexus, brain-cerebrospinal fluid barrier (BCB) and pia mater [23].

Ependyma is a monolayer of ciliated cuboidal ependymal cells lining the inner surface of cerebral ventricles and canalis spinalis that participates in establishing the essential exchange interface between brain interstitial fluid (BIF) and cerebrospinal fluid (CSF) [24]. A recent study showed that the neurotrophic factors and nutrients contained in the CSF from young mice improved the cognitive function of aged mice [25]. A decreased amount of these neurotrophic factors and increased accumulation of inflammatory factors in CSF may damage the normal structure and function of ependyma; disturb the formation, substance exchange and circulation of CSF; disrupt the homeostasis of brain microenvironment; and enhance neurodegenerative process [26,27]. Besides these changes of components in CSF, the neurodevelopmental chart for human lifespan shows dramatic increases in ventricular volume during aging after 60 years old, which is more significant in AD patients [28]. We speculate that the dysfunction of ependyma may lead to an imbalance of brain microenvironment and then initiate the process of neurodegeneration.

This article will review the physiological roles of ependyma in maintaining brain microenvironment, the pathological roles of dysfunctional ependyma in the pathogenesis of neurodegenerative diseases and RIBI, and the neurotrophic factors and signaling pathways influencing ependymal function. By maintaining the structure and function of ependyma at the early stage of neurodegenerative diseases, valuable therapeutic strategies could be designed to control the pathogenesis of those diseases and prevent dementia and dyskinesia.

## 2. The Physiological Function of Ependyma

Ependymal cells (ECs) line the ventricles and the central canal of the spinal cord [29], forming the brain’s ventricular epithelium and a niche for neural stem cells in the ventricular–subventricular zone (V-SVZ) [30]. These dormant stem cells can be elicited for differentiation and migration after activation [31]. A single-cell transcriptomic study has distinguished the ECs from the ependymal neural stem cells in the V-SVZ and verified no stem cell profile in ECs [32]. In the model of spinal cord injury, ECs showed a limited contribution in astrocytic scar-forming [33]. These findings challenged the hypothesis that the mature EC could function as a neural stem cell. Most mouse immature ECs are derived from radial glial cells around embryonic days 14–16, and then differentiated and matured with cilia formation in neonatal age [34,35]. There are three subtypes of ECs, multi-ciliated, bi-ciliated and mono-ciliated ECs [36]. The bi-ciliated and mono-ciliated ECs indicate the subtypes of tanycytes—the specified ECs. The coordinated beating of those propeller-like motile cilia protruded from ECs into the brain ventricles generates a directional CSF flow, which is essential for various physiological processes [37] (Figure 1A). To organize the formation of CSF, single-layer epithelial cells cover capillaries at the bottom of the lateral ventricle, the top of the third ventricle and the lower part of the fourth ventricle near the inferior medullary velum to form the choroid plexus [38] (Figure 1B). The stroma inside the choroid plexus is a part of the pia mater. As the wall of the ventricular system, ECs and astrocytes from the brain parenchyma form a well-controlled filtration membrane, the BCB, which promotes the bi-directional substance exchange between the BIF and CSF, keeps the brain tissue toxicant-free and in physiological balance [28,30,31,37,39,40] (Figure 1C).

Tanycytes are highly specialized ECs that play a vital role in forming the ependyma of the circumventricular organs (CVOs) [41]. Often described as “brain windows”, the CVOs are rich capillary networks closely contacted with tanycytes and continued with the neighboring choroid plexus. This unique structure allows a potential functional relationship of the capillary system with CSF. For example, the median eminence (ME) is a well-known CVO located in the tuberal region of the hypothalamus [42] (Figure 1D). Tanycytes are bi-ciliated or mono-ciliated ECs with less motility than multi-ciliated ECs, but have long processes that can across the hypothalamic parenchyma and link the ventricular and vascular compartments directly [36,43,44]. In the ME, the adjacent tanycytes adhere with each other by various tight junction proteins, including ZO-1, occludin, claudin 1 and claudin 5, to prevent the free passage of molecules through the paracellular pathway [45,46,47]. These tight junctions of tanycytes at the ventricular surface of the CVO can prevent the diffusion of blood-borne molecules into the CSF, even if those molecules have permeated into the parenchyma of the ME through the vasculature surface of the CVO. Tanycytes also take part in forming the BBB between the hypothalamic parenchyma and capillary to maintain the microenvironment surrounding those neuroendocrine cells and facilitate the release of hypothalamic regulatory peptides [43,44,47] (Figure 1D). In addition, the tanycytes may play a vital role in metabolic homeostasis by secreting or transporting circulatory fibroblast growth factor 21 (FGF21) into the central nervous system [48]. Unlike the stricter BBB formed intactly by astrocytes, tanycytes may provide a “window” for brain invasion while promoting substance exchange at the vasculature interface of the CVO [47,49].

The CSF produced in the brain ventricular system flows into the subarachnoid space through the median and lateral foramen of the fourth ventricle [50]. From here, the pia mater replaces the ependyma to form BCB. CSF circulating in the subarachnoid space drains into the subpial interstitial fluid (SPIF) from the perivascular spaces and exchanges substances with the BIF through the astroglial barrier or the blood through the capillary [51,52,53] (Figure 1E). Arachnoid granulations, which have been considered as the main pathway for absorbing CSF into venous sinus, may function as glymphatic–lymphatic coupling structures together with the newly unveiled subarachnoid lymphatic-like membrane (SLYM) [54,55,56]. The CSF-glymphatic communication through SLYM supervises the immune status of CSF and presents the information to the lymphatic and/or blood system through arachnoid granulations. Although the pia mater and ependyma develop from different origins, they both contribute much in maintaining the delicate balance of CSF dynamic flow and biochemical homeostasis in the brain microenvironment [57].

## 3. Ependymal Dysfunctions in the Pathogenesis of Neurodegenerative Diseases

Neurodegenerative diseases are associated with the abnormal transportation of metabolites or other substances among intracellular fluid, interstitial fluid, CSF and blood in the brain [58]. CSF is mainly produced in the choroid plexus and transported from the lateral ventricle to the third ventricle, aqueduct and fourth ventricle, and then is re-inhaled in the subarachnoid space [35]. As a fluid clearing pathway in the brain, the glymphatic–lymphatic pathway helps to drain the CSF from the subarachnoid space into the perivascular spaces of penetrating arteries, also known as Virchow–Robin spaces [59,60,61]. From these perivascular spaces, CSF can finally return to the brain parenchyma and/or the cerebral vasculature [59]. This lymphatic pathway dominates during sleep. It has been reported that the clearance rate of harmful metabolites (such as Aβ) during sleep was significantly higher than that during awaking [62,63]. During sleep, the BIF secreted from astroglia dilutes the extracellular metabolites and washes them away by the increased BIF advection in the larger interstitial space [64]. Recent research has demonstrated that the etiology of AD and other neurodegenerative diseases may involve the abnormal expression of lipoproteins from the reactive astrocytes, such as the intensively studied APOE4, and the neurotoxic lipids they transport [9,10,65]. These toxic lipids may disturb lipid metabolism in brain tissue and destroy the membrane structures, especially the ependyma. The tanycytes have been reported to have an important role in regulating lipid metabolism [66]. The aged mouse EC possesses more lipid droplet accumulation and loses its barrier function [67]. This metabolic alteration in EC can cause the aging of EC, the dysfunction of ependyma and cognitive impairment [68]. Neuron stem cell and other progenitors in the subependymal area, such as the SVZ, can repair the damaged ependyma; however, they sometimes induce gliosis on the surface of the ventricular wall [58,69,70]. The cross-talk between astroglial and microglia activation, perivascular macrophage migration and immune cell infiltration in SVZ after brain injury may affect periventricular interstitial fluid homeostasis and impair ependymal function [71,72].

The filtration of water through ependyma is mainly controlled by aquaporin 4 (AQP4), the most abundant aquaporin in the mammalian brain [38]. Increased AQP4 expression was detected at the gliosis site of ependyma that impaired the CSF/BIF dynamic balance and the clearance of interstitial solutes [58,73]. On the other hand, deletion of AQP4 can obviously prevent the cytotoxic edema after stroke [73,74]. The abnormal expression of AQP4 is also involved in the dysfunction of the lymphatic pathway in animal models of traumatic brain injury, AD and stroke [75]. A higher AQP4 level was found in the ECs after subarachnoid hemorrhage, and the expression level of AQP4 was related with the severity of hydrocephalus [76]. The autoimmune antibodies from the patients of neuromyelitis optica can target AQP4 on the surface of ECs to trigger the functional impairment and inflammatory response in ependyma [77]. There is no doubt that AQP4 variation is associated with genetic susceptibility to PD [78]. The choroid plexus epithelium also expresses other AQPs including AQP1, AQP5 and AQP7, which more or less contribute to CSF production [38].

The normal activity of the ependymal motile cilia ensures the necessary CSF circulation to maintain brain homoeostasis, wash out toxins, deliver signal molecules and orient the migration of new-born neurons [79]. However, the molecular mechanism underlying the maintenance of ependymal motile cilia remains unclear [80]. The highly conserved cilia project from the apical surface and the zonula adherens on the lateral surface of ECs to move the overlying fluid by coordinated beating [39]. These cilia arise from the basal bodies, which are docked on the cell surfaces and rotationally polarized toward the CSF [81,82]. It has been suggested that this complementary polarization of the ependymal cilia should be regulated by the planar cell polarity pathway, which coordinates cell behavior in a plane of tissue cells [83,84]. The motile cilia dyskinesia can cause chronic recurrent respiratory infections, infertility, hydrocephalus and laterality defect [85,86,87]. Defective ependymal cilia motility is associated with the hydrocephalus, increased intracranial pressure and many neurological diseases [88,89]. Ciliary defects in mouse ECs can disrupt the CSF flow and lead to hydrocephalus and disoriented neuroblast migration in the SVZ [90,91]. Connexin 43 (Cx43), the dominating connexin of gap junction in the brain, plays a vital role in maintaining ependymal cilia [29]. Deletion of Cx43 can reduce the ciliary activity of ECs in zebrafish and mouse [29]. Possibly, the absence of Cx43 may affect the polarization of the ependymal cilia through the planar cell polarity pathway.

The neurodegenerative diseases share similar changes in the brain at the early stage, such as hydrocephalus [92] and ventricular broadening [89]. To date, the final diagnosis of AD can only be made by histopathological detection of Aβ plaques and neurofibrillary tangles post mortem [93]. For the purpose of early diagnosis and prevention of AD, positron-emission tomography (PET) has been used to analyze the synaptic dysfunction and cerebral Aβ load in the brain of AD model mice [94]. The data indicated that the glucose metabolism was decreased and the Aβ deposition was increased in AD mouse brain. The decreased glucose metabolism in AD may be due to the dysfunction of those glucose transporters expressed in the BBB, choroid plexus and ependyma [95,96]. Interestingly, high glucose or fructose concentration can directly stimulate the expression of brain-derived neurotrophic factor (BDNF) in the mouse microglia SIM-A9 cell [97]. Besides Aβ accumulation, more activated microglia have also been reported in AD animal models and in patients [98,99]. The Aβ plaque can activate microglia to form the plaque-microglial complex, and then significantly alter the gene expression and biological function of the surrounding astrocyte and oligodendrocyte precursor cell [100]. Furthermore, vascular risk factors such as hypercholesterolemia and hyperglycemia may also be involved in the genesis of AD and other neurodegenerative diseases [62]. The severity of cerebral atherosclerosis and/or arteriolosclerosis are associated with cognitive dysfunction [101]. Improving Aβ clearance along the perivascular pathway may provide a feasible therapeutic approach to control the progression of AD [102]. Recent research demonstrated that the CSF macrophages near the border of brain parenchyma had a role in regulating CSF flow dynamics by delicate clearance of the extra accumulated extracellular matrix proteins [103]. The single-nucleus RNA sequencing data obtained from the AD patients and the animal model of AD demonstrated abnormal transcriptomic alterations in these macrophages [103]. Intracisternal injection of macrophage colony-stimulating factor can improve the function of CSF macrophages and restore the CSF flow, implicating a new strategy to counteract the deficient CSF dynamics [103].

Similarly, the early diagnosis of PD, especially the premotor phase, is difficult in a clinic setting. Intracellular accumulation of the α-syn aggregates is the major pathological change of PD [3]. A previous study demonstrated that the changes in sleep-related oscillations should be an early consequence of abnormal α-syn aggregation in the mouse model [104]. The lymphatic pathway helps to drain the harmful substances in the cerebral interstitial fluid and CSF through the perivascular spaces of penetrating arteries, especially during sleep [57]. Chronic sleep deprivation or circadian disruption may cause lymphatic pathway dysfunction in the brain. The consequent abnormal accumulation of α-syn or other harmful substances caused by this BCB dysfunction will consequently result in AD, PD, depression and anxiety [104,105,106,107].

HD is a genetic neurodegenerative disease caused by the abnormal expansion of the CAG trinucleotide repeat in the huntingtin gene, which leads to a polyglutamine strand at the N-terminus of huntingtin protein [108]. Current therapeutic strategies designed for HD focus on reducing cytoplasmic aggregation of the mutant huntingtin protein [16]. Most cases of ALS are also characterized by the abnormal cytoplasmic aggregation of different proteins including TAR DNA binding protein 43 (TDP-43), Cu–Zn superoxide dismutase (SOD1), ubiquitin/p62 and others [4,109,110]. Unlike the HD, many genetic mutants have been identified in the ALS patients. Therefore, it is complicated to explain the pathogenesis of ALS. Ageing or exogenous risk factors may accelerate these inherited sensitivities and cause the onset of neurodegeneration [4]. Without considering the initiation of neuron damage, the activation of microglia and astroglia may contribute to the progressive motor neuron loss in ALS [4,109]. In human HD brains, the inflammatory activation of astroglia in the caudate nucleus and the subependymal layer was indicated by the co-localization of RAGE with its ligands and the nucleus translocation of NF-κB [111].

Under most circumstance, preventing neuroinflammation at an early stage can improve the cognitive impairment [112,113]. However, inhibition of the proinflammatory kinase IKKβ accelerates HD progression in mice because IKKβ has a role in phosphorylating huntingtin [5]. A review article suggested that the activation of microglia and astroglia in brain tissue may promote the BBB restoration, limit the blood-derived immune cell infiltration, trap the infiltrated T cells and achieve the early resolution of neuroinflammation in multiple sclerosis [114]. The age-related cerebral microvascular dysfunction and microbleeding destroy the integrity of BBB and allow the entry of peripheral neurotoxic substances, macrophages and neutrophils [115,116,117,118,119,120]. These factors can activate microglia and astroglia in the brain to release pro-inflammatory cytokines that induce chronic neuroinflammation and further brain injury. The anti-inflammatory reagent OKN-007 has shown effects on reversing lipopolysaccharide (LPS)-induced long-term neuroinflammatory responses and BBB impairment [121].

## 4. Ependymal Dysfunctions in RIBI

As a protective barrier lining the brain ventricles, the ependyma is extremely vulnerable to cytotoxic and cytolytic immune responses. The low-dose ionizing radiation showed obvious benefits on the cognition and behavior of severe AD patients in a clinical trial [18]. It is supposed that the radiation impairs the BBB and other brain barriers to facilitate the CSF dynamic flow and substance exchange [17]. Epidemiological studies have shown that the ionizing radiation exposure from the nuclear incident and radiotherapy may cause acute brain damage and chronic alteration of the neurons, glia and cerebrovascular endothelium [122]. Radiation-induced BBB disruption, microglia activation and neuroinflammation may damage the ependyma and disturb the neurogenesis and/or gliogenesis in the SVZ [123]. Our previous data have suggested that some herbal drugs prevent neuron loss and cognitive impairments probably by stimulating neurogenesis and eliminating the neuroinflammatory factors and cytotoxic metabolites [124,125]. In the RIBI mouse model, pregabalin can reduce neuron loss by inhibiting microglia activation [20]. In the RIBI patients who received head and neck radiotherapy, the activated microglia recruit cytotoxic CD8+ T cell infiltration through the damaged BBB [21]. A clinical trial indicated that the thalidomide restored the BBB in some RIBI patients and improved their cognitive function [126]. Although the dysfunction of ependyma has not been mentioned directly in previous studies of RIBI, it is reasonable to assume that the inflammatory disruption of BCB and other brain barriers should also contribute to the chronic cognitive decline after RIBI and other acute brain injuries.

Besides the local microglia, the infiltrated cytotoxic immune cells [21] and the autoimmune antibodies [77], the complement proteins can immediately participate in the identification, transportation and removal of pathogens and unwanted host substances [127]. The production and activation of complement proteins are the important parts of the innate immune response in the body fluid and the tissues [128]. The complement proteins in CSF are probably synthesized by neurons, microglia, astrocytes and oligodendrocytes, especially during brain infection or neuroinflammatory diseases [129]; they have the potential to activate downstream mediators and thereby promote the further inflammatory response. For example, the activation of complement 3 (C3) can induce the release of tumor necrosis factor (TNF), which in turn leads to the activation of interleukin-1 (IL-1), IL-6 and IL-18 [129,130]. The involvement of Aβ plaque in the complement activation has been reported since the 1980s [131]. Studies have shown that the specific assembling of Aβ could activate the complement pathway in vitro through the interaction with C1q, a subunit of C1, or directly with C3 [132,133,134]. The activated complements such as C3a and C5a have a chemotactic effect on microglia and astrocytes [135,136]. They recruit microglia and astrocytes around the Aβ plaques to further activate the pro-inflammatory factors, reactive oxygen species and proteases, which may accelerate neuronal dysfunction and cognitive dysfunction [137]. In addition, the C1q mediates the phagocytosis of glutamatergic synaptic microglia during the pathogenesis of AD [138,139]. The dysfunction of glutamate transporter 1 (GLT1) may induce the synaptic and cognitive dysfunction in AD through the C1q-mediated phagocytosis of synaptic microglia [140]. As the initiator of the classical pathway of complement activation, the expression of C1q is up-regulated in neurons and glial cells in many neurodegenerative diseases such as AD, PD, HD and frontotemporal dementia [141,142,143,144].

In the acute phase of meningitis, the complement fragments and other immune factors in CSF can promote the infiltration of phagocytes and destroy the BCB by forming the cytotoxic and cytolytic attack complexes [145]. In neuromyelitis optica cases, more deposition of C9neo (a complement attack complex), microglia activation and granular cells infiltration, and less AQP4 expression, were observed in the pia, ependyma and choroid plexus epithelium when compared to the normal and multiple sclerosis tissues [146]. However, the ependymal layer and choroid plexus can express the membrane-bound complement regulators (CRs) to inhibit C3 convertase (CR1/CD35; membrane cofactor protein, MCP/CD46; decay accelerating factor, DAF/CD55) or avoid the formation of the membrane attack complex (CD59) in vitro and in situ [145]. The level of these membrane-bound CRs (CD35, CD46, CD55 and CD59) on the choroid plexus epithelial cells and ependyma are significantly increased in meningitis [145]. Therefore, it is proposed that the activation of the complement system is the major cause of ependymal damage during neuroinflammation—even the ECs can protect themselves by expressing CRs.

In summary, neuroinflammation can be caused by many reasons including brain infection, acute neuron loss of RIBI, metabolite aggregation, autoimmune diseases and others. The activation of the complement system may indicate the inflammatory status of ependyma together with other factors. As discussed above, BCB dysfunction can be caused by the neuroinflammatory factors, cerebral microvascular disorders, circadian disruption and others. The dysfunction of ependyma, the most important layer of BCB, may largely be involved in BCB dysfunction and lead to the imbalance of brain microenvironment. Similarly, the onset of other neurological diseases may also involve the dysfunction of BCB and the imbalance of cerebral microenvironment, such as the abnormal synaptic activity in epilepsy [147]. As summarized in Figure 2, inflammation induced complements and other inflammatory factors, such as the autoimmune antibodies, infiltrating into and circulating in CSF can attack ependymal cells and consequently lead to less ependymal motile cilia or loss of ependymal integrity. The direct exposure of the sub-ependymal area to CSF may promote the recruitment of monocytes and the maturation of macrophages. The activated macrophages may enhance the ependyma injury while promoting the clearance of debris. At the same time, the ependyma impairment may activate astroglia and induce the activation of microglia locally, which may damage neurons and oligodendrocytes to present autoantigens, as we have observed in many neurodegenerative diseases. The immediate restoration of ependymal integrity may more-or-less restrain the exposure of these autoantigens and the consequent autoimmune responses. However, the formation of astroglia scar in the ependymal layer may reduce the permeability of the barrier and cause the accumulation of neurotoxic debris and metabolites in the brain tissue. Therefore, reviewing the possible neurotropic factors and regulatory pathways, which can maintain or restore the ependymal function, may implicate the alternative therapeutic targets of the neurodegenerative diseases, especially at the early stage.

## 5. Neurotropic Factors and Regulatory Pathways Maintaining Ependymal Function

The transcriptomic analysis shows that the ECs express several stem-cell-related genes, suggesting that the ECs may be potential neural stem cells that can proliferate and differentiate into scar-forming astrocytes after brain injury [31,32]. However, both in vitro and in vivo studies indicate no correlation of ECs with neural stem cell or progenitors [32]. The ciliated mature ECs may have limited proliferative potential to regenerate themselves or act as their precursor radial glial cells to mainly regenerate astroglia [33,148,149]. The induced expression of oligodendrocyte lineage transcription factor OLIG2 in ECs triggered their differentiation into oligodendrocytes, implicating the inducible neurogenesis potential of ECs [148]. Brain-injury-induced responses, such as the activation of microglia, may promote the ECs differentiate into scar-forming astroglia [149]. When the ependyma was impaired in the mouse model of neuroinflammation, neural stem cells proliferated rapidly to produce astroglia and a small number of oligodendrocytes but not neurons [150]. At a molecular level, the transcriptional factor FoxJ1 may be an essential factor for the differentiation of radial glial cells into ECs because of its role in ciliogenesis [151]. Poorly differentiated ependymomas express much less FoxJ1 and ciliogenesis-related genes [152]. Nuclear factor IX may interact or regulate FoxJ1 to affect the differentiation and maturation of ECs [153]. Matrix Metalloproteinases (MMPs), a family of zinc-dependent endopeptidases, can be divided into two types, i.e., membrane and secretory types. The former is localized in the cell membrane and the latter is secreted into extracellular fluid [154]. They function to degrade the extracellular matrix. The dysfunction of MMPs has been involved in a variety of neurological disorders, such as AD, PD, multiple sclerosis and glioma [155]. It has been shown that MT1-MMP is highly expressed in ECs lining the ventricles, and loss of MT1-MMP results in impaired EC maturation [154]. Transcriptomic analysis indicates higher involvement of the metal-related genes than cilia-related genes in the ECs lining adult human and mouse brain ventricles, implicating the essential role of the mature ECs in metal ion transportation and extracellular matrix metabolism [156]. In addition, the dysfunction of vacuolar protein sorting-associated protein 35 (Vps35), a key component of the retromer complex, is believed to be a risk factor of neurodegenerative disorders including AD and PD [157,158]. The Vps35 is highly expressed in developing ECs, and the knockout of Vps35 leads to the reduction in ECs and their motile cilia [159]. In multi-ciliated ECs, the planar cell polarity coordinates their ciliary pulsation and direct CSF circulation. p73 can regulate ependymal planar cell polarity by modulating the assembling of microtubule, especially during the postnatal period [160,161]. Although the deletion of p73 in adult ECs did not affect the maintenance of EC polarity, the loss of p73 during the developing stage of ECs led to a later aqueduct stenosis and the development of hydrocephalus [161]. Upregulation of the expression of these EC-specific genes, especially FoxJ1, may promote the generation and maturation of ECs and keep the integrity of ependyma.

Kojima et al. also demonstrated that the continuous intrathecal administration of epidermal growth factor (EGF) and fibroblast growth factor 2 (FGF2) significantly stimulated the proliferation of ependymal precursor cells and expansion of ECs and astrocytes in adult rat spinal cords [162,163]. O’Hara and Chernoff found that growth factors modulated injury-reactive ependymal cell proliferation and migration [164]. Their results indicated that the EGF promoted ependymal migration and proliferation in vitro, while platelet-derived growth factor (PDGF) and transforming growth factor-beta 1 (TGF-β1) showed no or inhibitory effect on ependymal proliferation but affected the reorganization of cultured ECs. TGF-β1 can inhibit the ciliogenesis and maturation of ECs in the polarized primary culture system [165]. In the rat model of chronic hypoxia, the increased expression of FGF2 and its receptor in the SVZ was observed [166]. It is suggested that the FGF receptor signaling in radial glia close to those FGF2-positive cells in the SVZ might promote their proliferation and differentiation into astroglia and ECs. The ependyma may also have FGF-sensitive progenitor cells for neurogenesis [167]. Intraventricular administration of insulin-like growth factor 1 (IGF-1) can stimulate significant proliferation of neurons, astrocytes, tanycytes, microglia and endothelial cells, but not ciliated cubic ependymal cells, in the periventricular and the parenchymal zones of the whole hypothalamus [168]. The platelet-derived growth factor receptor beta (PDGFRβ) is expressed by the neural stem cells in the V-SVZ [169]. Selective deletion of PDGFRβ in these stem cells may disinhibit them, leading to the changes in their status from the quiescent to active one, i.e., gliogenesis. The proinflammatory cytokine interleukin-17A, by binding with its receptor expressed on the surface of EC, down-regulates the transcriptional levels of neurotrophic factors including nerve growth factor (NGF), glial cell-derived neurotrophic factor (GDNF), ciliary neurotrophic factor (CNTF), PDGF, TGF-β1 etc. [170]. The anti-inflammatory strategy may facilitate the release of endogenous neurotrophic factors. Until now, it seems that the EGF is the most acceptable growth factor that can benefit the integrity of the ependyma. However, the detailed mechanism and signal pathways underlying EGF-mediated proliferation of ECs are still unclear.

Intraventricular injection of EGF induces continues extracellular-signal-regulated kinase (ERK) and cyclic AMP response element-binding protein (CREB) activation in the progenitor cells and ECs, which may contribute to the intense periventricular cell proliferation [171]. The activation of ERK/CREB pathway may affect the downstream expression of Wnt family members, which are important mediators of numerous developmental events [172]. Other indirect evidences have demonstrated that the Wnt signaling pathway is involved in the development of ependyma and its motile cilia [173,174,175,176]. Wnt molecules are well known for their roles in embryogenesis, carcinogenesis, cell proliferation and differentiation [177,178,179]. The presence of Wnt molecules in the tanycytes and ECs lining the third ventricle and arcuate nucleus of the hypothalamus may regulate the ependymal function through β-catenin dependent signal pathway, c-jun N-terminal kinase (JNK) or Ca^2+^ dependent signal pathway [177]. Many Wnt proteins (Wnt1, Wnt3a and Wnt10b) can activate β-catenin, whereas only a few Wnt proteins (Wnt5a, Wnt11) are involved in the β-catenin independent pathways [177]. In the Wnt/β-catenin signaling pathway, the Wnt binds with a frizzled (fzd) receptor and then forms a complex with the co-receptor lipoprotein-receptor-related protein (LRP) to facilitate the disheveled (Dvl) dependent phosphorylation of LRP, which consequently inactivates the glycogen synthase kinase 3β (GSK3β) [177]. GSK3β inactivation decreases the phosphorylation level of cytoplasmic β-catenin, and the stabilized β-catenin can enter the nucleus to regulate the transcription of those downstream target genes such as cyclin D1 and Axin2 [177,180,181].

The ECs in postnatal and adult spinal cord express the Wnt proteins and the Wnt/β-catenin signaling target gene Axin2 [174,175]. Wnt/β-catenin signaling activity, indicated by the expression of Axin2, has been detected in the ECs and the neural progenitor cells, which can differentiate into ECs in the postnatal and adult spinal cord [175]. The proliferation rate of Axin2+ ECs in the β-catenin knockout mice was significantly lower than that of Axin2+ ECs in the age-matched control group during both postnatal and adult ages [173]. The results of vimentin and Ki67 immuno-staining demonstrated that the number and proliferation of ECs decreased when the expression level of the Wnless (a Wnt-specific transporter) gene in Axin2+ ECs was inhibited [175]. These results suggest that the Wnt/β-catenin signaling pathway is necessary to promote EC proliferation and maintain ependyma integrity after spinal cord injury.

The Wnt signaling pathway also plays a role in the development and maintenance of ependymal motile cilia [29,176]. The expression of Wnt4b, Wnt5b, Wnt11 [182,183] and fzd7b [184] have been identified in the ECs. Suppressing the Wnt signaling by using Dickkopf 1 (Dkk-1) [183] or IWR-1 [185] diminishes the development of ependymal motile cilia in the wild-type zebrafish embryos. The same result has been observed in the fzd7b knockdown embryo [184]. During the development of ependymal motile cilia, the asymmetric localization of PCP proteins is required for the dynamic planar polarization of microtubules, which coordinates cilia orientation [83]. The Daple protein could regulate the dynamic apical and planar polarization of the microtubules during ependymal development by interacting with Dvl [186,187]. However, the assembly of basal bodies in ECs was not affected as much as that of motile cilia when the embryo was treated with IWR-1 [185]. It seems that the defect in the Wnt signaling pathway may not affect the development of motile cilia through the PCP pathway [188]. The most possible mechanism could be that the Wnt signaling pathway affects the development and function of ependymal motile cilia through regulating intercellular Ca^2+^ wave (ICW) propagation [29,154,189,190]. The Wnt signaling may promote the expression of PLCδ3a and enhance the expression of Cx43 to regulate ICW propagation among ECs through gap junctions [85,86,87,189] and then coordinate motile ciliary beating [190]. Wnt/β-catenin may also up-regulate FoxJ1 expression and ciliogenesis in the zebrafish Kupffer’s vesicle [191].

The inhibition of GSK-3 and the associated activation of Wnt/β-catenin signaling could increase astroglial migration, reduce astroglial apoptosis and enhance axonal growth to promote motor function recovery following the spinal cord injury [192,193]. The GSK-3 inhibitor Ro3303544 can activate β-catenin signaling and increase the number of terminally differentiated neurons in the cultured ependymal stem/progenitor cells (epSPCs) in mouse spinal cord [194]. In the Ro3303544 treated mouse model of severe spinal cord injury, the numbers of bromodeoxyuridine (BrdU) and doublecortin (DCX) positive neurons increased, the area of astrocyte scar boundary decreased and the motor function was improved [194]. The acute transplantation of epSPCs after spinal cord injury can improve the functional locomotor recovery and increase the expression of the purinergic P2X4 and P2X7 receptors, which are regulated by Specificity protein 1 (Sp1) transcription factor [195,196]. The Sp1 factor plays a key role in initiating the transcriptional mechanism involving GSK3 and other intracellular kinases to induce the expression of regeneration-related genes [196]. These results implicate that the Wnt signaling pathway may be a valuable target for improving the development, integrity and function of ependyma (Figure 3).

## 6. Conclusions

Ependymal cells serve important sensory and mechanical functions, including signal transduction, movement of CSF and insulation of neural tissues from peripheral hazardous substances [197]. Abnormal changes of the ependyma may affect the brain microenvironment, leading to the accumulation of harmful substances and cerebral spinal fluid flow disorder. Ependyma acts as a physical barrier in the BCB, affecting the exchange of CSF and BIF. The movement of ependymal cilia promotes the flow of cerebrospinal fluid, which is associated with the removal of metabolites or waste products in the brain. Extensive studies have indicated that the abnormal changes of ependyma may result in the imbalance of the brain microenvironment, which may be related to different neurodegenerative diseases [62,99,198]. Targeting abnormal brain ependymal changes may prevent brain microenvironment changes such as accumulation of metabolic waste in the interstitial fluid, glial activation, inflammation and subsequent neurodegenerative disease genesis. Neurotrophic factors or growth factors, such as EGF and FGF2, may stimulate the proliferation and differentiation of sub-ependymal neural stem cells into astroglia and ECs, restore the ependyma and maintain the microenvironmental homeostasis of the central nerve system, suggesting that the ependyma may be a valuable therapeutic target to prevent imbalanced-microenvironment-related neurodegenerative diseases.

It must be emphasized that our current understanding of the roles of ependyma in the normal brain and pathophysiological changes of the brain is still very limited. The direct link of the pathological changes of ependyma to each neurodegenerative disease remains unclear. The long-term effect of radiation damage on the ependyma still needs to be investigated. The therapeutic effect of neurotrophic factors or growth factors on maintaining or restoring ependymal function during the pathogenesis of neurodegenerative diseases or after irradiation also needs to be studied further. Whether neurotrophic factors or growth factors initiate or promote the expression of FoxJ1 in the neural stem cell or progenitor cell to induce the proliferation, differentiation and maturation of ECs remains to be elucidated.

## Figures and Tables

**Figure 1 biomolecules-13-00754-f001:**
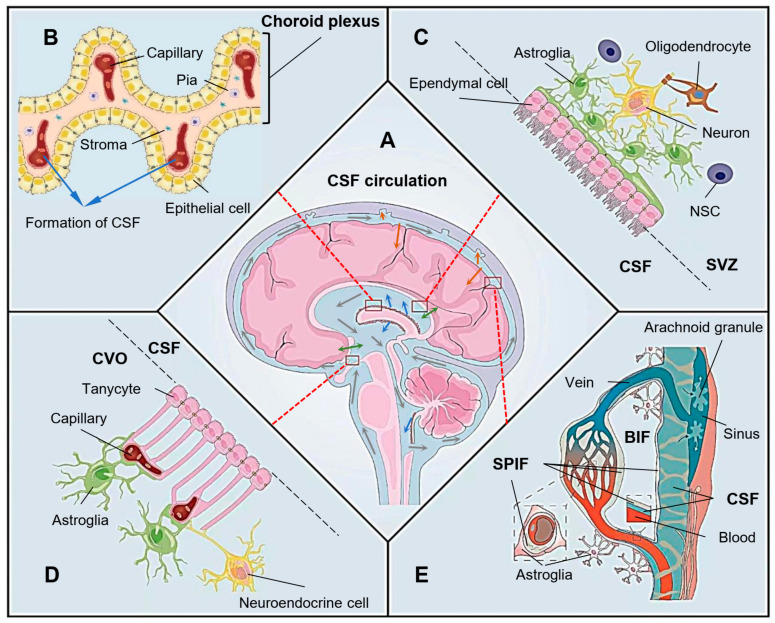
Role of the ependyma in cerebrospinal fluid (CSF) flow dynamics. (**A**) Generation, substance exchange, circulation and draining of CSF. The blue arrows indicate the generation of CSF from the choroid plexus; the grey arrows indicate the direction of CSF flow through the brain ventricular system and subarachnoid space; the green arrows indicate the substance exchange of CSF and brain interstitial fluid (BIF) or blood at the ventricular wall; the orange arrows indicate the draining of CSF at the perivascular spaces and arachnoid granules. (**B**) The formation of CSF from the choroid plexus, which is formed by a single layer of epithelial cells; the stroma derived from pia mater; and the capillary endothelium. (**C**) Ependyma and astroglia form the ventricular wall, which functions as the brain–CSF barrier. The neural stem cell (NSC) located in the niche of subventricular zone (SVZ) may proliferate to repair the damaged ependyma and regenerate astroglia to restore the barrier. (**D**) Tanycytes interact with astroglia and blood vessels to form a three-directional interface facilitating substance exchange among CSF, BIF and blood. These “brain windows” of the circumventricular organs (CVOs) play a vital role in the transportation of hypothalamic regulatory peptides and other factors. (**E**) CSF flows into the perivascular space and drains into the subpial interstitial fluid (SPIF), which can exchange with the BIF through the astroglial barrier and the blood through the endothelium. The CSF here also acts as a glymphatic system to introduce immune supervision and facilitate waste clearance from the BIF. Arachnoid granules may function as the location for “dirty” CSF draining back into the vein sinus or lymphatic pathway.

**Figure 2 biomolecules-13-00754-f002:**
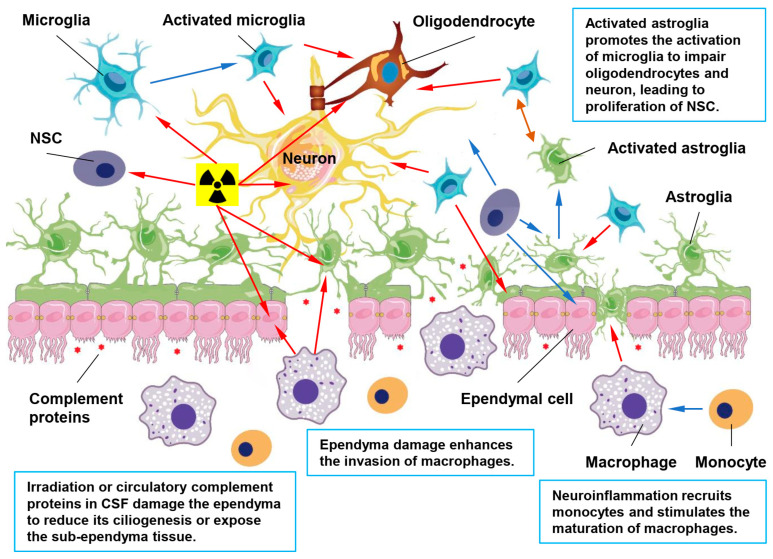
Ependymal dysfunction in neurodegenerative diseases. The ependyma can be damaged by the radiation exposure, cerebrovascular events, inflammation induced complement proteins, recruited perivascular macrophages and even the activated astroglia and microglia from the sub-ependyma region. Brain injury may activate the silence neural stem cell (NSC) to proliferate and differentiate into astroglia to repair the damaged ependyma. This glia-scar in the ependyma mediates neuroinflammation and restrains trans-barrier substance exchange, leading to an imbalance of the brain microenvironment. Under these circumstances, continuous neuron loss, abnormal neurogenesis and gliogenesis may lead to neurodegenerative diseases. Blue arrows show the proliferation, differentiation, maturation and activation processes. Brown arrows shows the crosstalk between activated astroglia and microglia. Red arrows indicate the damages of the neuron, glia, NSC and ependymal cell.

**Figure 3 biomolecules-13-00754-f003:**
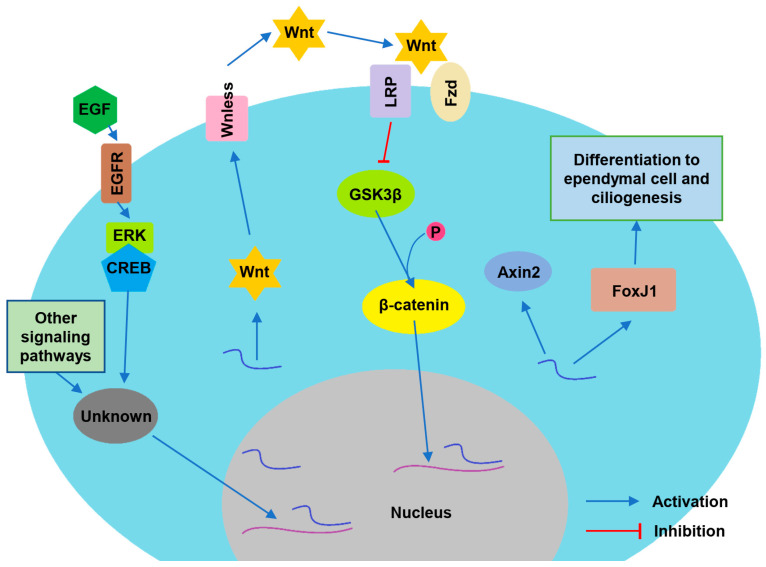
Signaling pathways involved in the maturation and ciliogenesis of ependymal cell. EGF and other growth factors or neurotrophic factors may enhance the transcription of Wnt proteins through ERK/CREB or other signaling pathways. Wnless promotes the secretion of Wnt, which binds with the frizzled (Fzd) receptor to activate LRP. The phosphorylated LRP inhibits GSK3β mediated phosphorylation of β-catenin. Without phosphorylation, the stable β-catenin enters the nucleus to promote the transcription of Axin2, FoxJ1 and other molecules that may facilitate the differentiation to ependymal cell and ciliogenesis.

## Data Availability

No new data were created or analyzed in this study. Data sharing is not applicable to this article.

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
