# Peer review of "Ependyma in Neurodegenerative Diseases, Radiation-Induced Brain Injury and as a Therapeutic Target for Neurotrophic Factors"

_biomolecules, 2023, doi:10.3390/biom13050754_

Round 1

Reviewer 1 Report

Overall, the authors highlight the immune/inflammatory-mediated mechanisms of ependymal cell dysfunction in neurodegenerative disease, and it is interesting how neurotrophic factors and growth signaling pathways may restore ependymal cell functionality.

Suggestions changes to improve paper:

Missing in-text citation for lines 41 and 42 (World Health Organization Statistics)

There are 3 subpopulations of ependymal cells that have been previously described – E1 (multi-ciliated), E2 (bi-ciliated), E3 (uni-ciliated) ependymal cells. All 3 subpopulations should be mentioned and described, not just 2 of them.

In lines 76-78, the authors are describing the CSF-brain barrier. This term should be explicitly mentioned in this section, since it is mentioned in the legend for Figure 1C which corresponds to these lines. Ependymal cell dysfunction within the CSF-brain barrier (and not just the blood-CSF barrier) should be further described.

Tanycytes also form components of the spinal ependymal cell layer. The spinal ependymal cell layer should be mentioned and described for completion.

Line 125 – line should read, “…during sleep may be due to the larger volume…”

Line 126 – line should read “recent research” instead of “recent researches”

It is unclear what is meant by “deterioration” in Lines 131-133: “The dysfunction of ependyma will deteriorate the brain injury and inflammation induced astroglial and microglia activation, perivascular macrophage migration and immune cell infiltration, and then affect the periventricular interstitial fluid homeostasis”.

Regarding the section on Alzheimer’s disease, altered lipid and glucose metabolism in ependymal cells should be described in more detail.

Line 223 – line should read, “The direct exposure of sub-ependymal area…”

Include a brief section on ependymal cell dysfunction in Huntington’s disease and amyotrophic lateral sclerosis, as these are mentioned in the introduction as common neurodegenerative diseases and there are studies of ependymal cell dysfunction in these diseases as well.

The potential role of neurotrophic factors, growth factors and Wnt/B-catenin signaling pathways in restoring ependymal cell functionality are well described and organized. The diagram (Figure 3) is informative and well-illustrated.

In general, there could be more of an emphasis on how radiation exposure influences ependymal cells. A specific section dedicated to the effects of radiation on ependymal cells would be helpful, similar to how there is a specific section dedicated to ependymal cell dysfunction in neurodegenerative disease. Since radiation exposure appears in the main title, it should be more extensively covered.

Overall, the writing style, grammar, and flow could be refined.

Reviewer 2 Report

The manuscript represents a potentially very interesting review article, however some crucial mistakes were detected in the text which evoke significant doubts regarding the readiness of the article for publication

The description of neurodegenerative diseases including pathomechanisms and clinical presentations is oversimplified and thus false in general understanding. The topic of neurodegeneration is very complicated and far from well known. Many aspects are controversial and doubtful. Thus the authors cannot present general and final conclusions based on single reports even these very well published. Additionally, the provided information is not always consistent with the cited sources – e.g. the authors wrote that “ECs have been proposed as adult neural stem cells that provide the majority of newly proliferated scar-forming astrocytes that help to protect the neuronal function after spinal cord injury” BUT the cited publication reports that “ependyma ARE NOT A MAJOR SOURCE” of endogenous neural stem cells or neuroprotective astrocytes after SCI”.

Not only copied the authors virtually word for word the first line of the Abstract but they did not acknowledge the rest of it and this is not an example of sound scientific work…

The authors should check carefully the information provided in their manuscript for such fundamentally false information and submit the potentially interesting paper once more.

Reviewer 3 Report

Overall, the review is really interesting. They discuss the important role EC could have in neurodegenerative diseases and how their "stemnness" can be leveraged for therapy.   There are some minor points in the attached PDF.

Round 2

Reviewer 2 Report

The authors followed the suggestions of the reviewers, no further remarks.

Author Response

Thanks for your review.